# Genetic diversity of Ethiopian cocoyam (*Xanthosoma sagittifolium* (L.) Schott) accessions as revealed by morphological traits and SSR markers

Eyasu Wada[1]*, Tileye Feyissa[2], Kassahun Tesfaye[2], Zemede Asfaw[3], Daniel Potter[4]

1 Department of Biology, College of Natural and Compuational Sciences, Wolaita Sodo University, Wolaita Sodo, Ethiopia, 2 Department of MCMB, College of Natural and Computational Sciences, Addis Ababa University, Addis Ababa, Ethiopia, 3 Department of Plant Biology and Biodiversity Management, College of Natural and Computational Sciences, Addis Ababa University, Addis Ababa, Ethiopia, 4 Department of Plant Sciences, College of Agricultural and Environmental Sciences, University of California-Davis, Davis, California, United States of America

* eyasuwada@gmail.com

**Data Availability Statement:** All relevant data are within the manuscript and its Supporting Information files.

## Abstract

Cocoyam (*Xanthosoma sagittifolium* (L.) Schott) is an exotic species from tropical America that is widely cultivated in Ethiopia for its edible cormels and leaves. There is a dearth of information on the genetic diversity of Ethiopian cocoyam. In order to evaluate and select cocoyam germplasm for breeding and conservation, genetic diversity of 100 Ethiopian cocoyam accessions (65 green- and 35 purple- cocoyam) were analyzed using 29 morphological traits (16 qualitative and 13 quantitative) and 12 SSR loci. Two classes of qualitative traits were observed. ANOVA revealed significant variation in 11 (84.6%) of the 13 studied quantitative traits. The SSR marker analysis showed high genetic diversity. A total of 36 alleles were detected with a range of 2 to 5 (average of 3.273) alleles per locus. The average observed heterozygosity (Ho) and expected heterozygosity (He) values across populations were 0.503 and 0.443, respectively. The analysis of molecular variance showed that the variation among populations, among individuals within populations, and within individuals explained 14%, 18%, and 68% of the total variation, respectively. Cluster analysis grouped the accessions irrespective of the collection sites. A dendrogram based on Nei's standard genetic distance grouped the green cocoyam accessions together while the purple cocoyam accessions occupied a separate position within the dendrogram. Significant variation in quantitative traits and the high level of genetic diversity revealed by the SSR markers suggest that diverse cocoyam accessions, probably with multiple lineage, were introduced multiple times, through multiple routes and probably by multiple agents, an hypothesis that needs futher testing and analyis. The crop, therefore, needs more research efforts commensurate with its economic and social values than it has been accorded thus far. Further study is recommended to clarify the taxonomic status of Ethiopian cocoyam accesions and to trace their evolutionary relationships with *Xanthosoma* species elsewhere.

**Funding:** This study was supported by funding to Eyasu Wada from the DAAD in country/in region scholarship 2015 (Personal ref. no.: 91602027), and partly supported by funding to Daniel Potter from the USDA National Institute of Food and Agriculture, Hatch project number CAD-PLS-6273-H. The funders have no role in study design, data collection and analysis.

**Competing interests:** The authors have declared that no competing interests exist.

## 1. Introduction

Cocoyam (*Xanthosoma sagittifolium* (L.) Schott) is a herbaceous, monocotyledons crop in the Araceae family. Cocoyam originated and was first domesticated in tropical America where it was cultivated from very ancient times [1]. It is one of the oldest cultivated aroids in the world [2]. Nowadays, it is cultivated in many tropical countries for its edible cormels and leaves [3].

There is much confusion and many discrepancies and uncertainties regarding the taxonomy of *Xanthosoma* at the species level [4]. Many names are associated with the cultivated species of the genus *Xanthosoma* [5, 6] but the name *Xanthosoma sagittifolium (*L.) Schott is generally given to all cultivated taxa of the genus [6]; several common names are also used to refer to the cultivated species [5, 7–10]. In this paper, cocoyam is refered to as *X. sagittifolium*, which is an accepted active name of the cultivated species of the genus *Xanthosoma* in the Araceae family [11] and in The Plant List [12].

Cocoyam was introduced into Eastern Africa through Western Africa [13]. In Ethiopia, cocoyam is largely unknown or considered synonymous with another related aroid, *Colocasia esculenta* (L.) Schott, which is locally known as taro or "*Godere*" [14]. However, cocoyam can be distinguished morphologically from taro by the place where the petiole meets the leaf blade. In cocoyam, the petiole is attached at the basal margin of the blade between the lobes. In taro, the petiole is attached in a peltate position, at or around the middle of the abaxial surface of the blade [5].

Cocoyam is a moneoecious species that flowers rarely. The unisexual flowers are born in a compound spadix with female flowers at the base, and the male flowers at above. Sterile flowers are located between the pistillate and staminate flowers. The inflorescence is protogynous; the stigma is normally receptive two to four days before pollen is shed [15], which makes classical breeding difficult. Cocoyam rarely sets sexually produced seeds and hence it is mainly propagated vegetatively from corm sets, headsets or cormels [16] but treatments were successfully applied to induce flowering with artificial pollination techniques. Hand pollination and production of seed and seedlings have become possible in experimental settings [17, 18].

Cocoyam grows on a variety of substrates and in habitats ranging from full sun to deeply shaded areas under the canopy of natural forests. It tolerates drier, but not waterlogged soils [7]. It is a lowland plant but grows in highlands [15]. It is a perennial plant but most often cultivated as an annual crop for food value. Harvest occurs during the dry season, 9–12 months after planting [9]. At the end of the growing season, leaves die and shoots wither completely [16, 19].

Many developing countries in the tropics depend on aroids in the tribes *Colocasieae* and *Caladieae*, i.e., taro (*C. esculenta*) and cocoyam (*X. sagittifolium*) as sources of carbohydrate [20]. Cocoyam has overtaken taro as the main edible aroid in many tropical areas [3]. It is an important crop in many parts of the tropical world, mainly for smallholder farmers, in playing a major role in the lives of many as a food security crop. Both its leaves and its starch-rich cormels are eaten after cooking [2, 3].

In Ethiopia, cocoyam has spread widely and become an important part of the agriculture and food systems of the local people. There is a considerable amount of the cocoyam germplasm in farmers' fields [21–28] but the species has not received thoughtful research efforts as it is a neglected crop by research and the development community. The germplasm in the farmers' field is the major hope for use, conservation and improvement of the crop. Thus, collecting accessions and assessing genetic diversity are very important to identify material for conservation and efficient and effective usage in crop improvement.

Characterization of morphological traits is an essential step for effective utilization of germplasm because it offers a useful approach for assessing the extent of genetic diversity among

accessions [29, 30]. The genetic information provided by morphological traits is, however, often limited and expression of quantitative traits is subject to strong environmental influence [31, 32]. Assessing genetic diversity of plant species by using molecular markers has several advantages over characterization based on morphological traits [33] although the former cannot completely replace the latter. This study employed characterization using morphological descriptors and SSR molecular markers to assess the genetic diversity of Ethiopian cocoyam (*Xanthosoma sagittifolium* (L.) Schott) accessions to enhance effective usage, conservation, and improvement of the crop.

## 2. Materials and methods

### 2.1 Plant materials

Similar sized cormel (germplasm) of 100 cocoyam accessions were sampled from 5 zones (Bench-Maji, Kefa, Dawuro, Wolaita and Gamo-Gofa) (Fig 1). Twelve to twenty-eight accessions were sampled from each zone (S1 Table). The individual accessions were collected from at least 5 km apart unless they were clearly distinguished by leaf and petiole color difference (65 green- and 35 purple-cocoyam). Accessions collected from the same zone were considered one population for the genetic diversity studies based on the assumption that cormel germplasm was more likely shared within zones than among zones. The same language is spoken within each administrative zone with frequent local markets where local people buy and sell commodities. The accessions are kept as living material at the Areka Agricultural Research Center.

### 2.2 Experimental design and crop management

Cormel germplasm entries (100) were planted in a common research garden at Areka Agricultural Research Center, 303 km southwest of Addis Ababa, Ethiopia in a 10*10 simple lattice square design in well-drained, loose soil on flat ground in the 2016 cropping season. Five cormels of an accession were planted in a single row plot of 3 m, spaced 0.6 m between plants and 0.75 m between rows. Weeding was conducted as required to keep plots weed-free. Similar

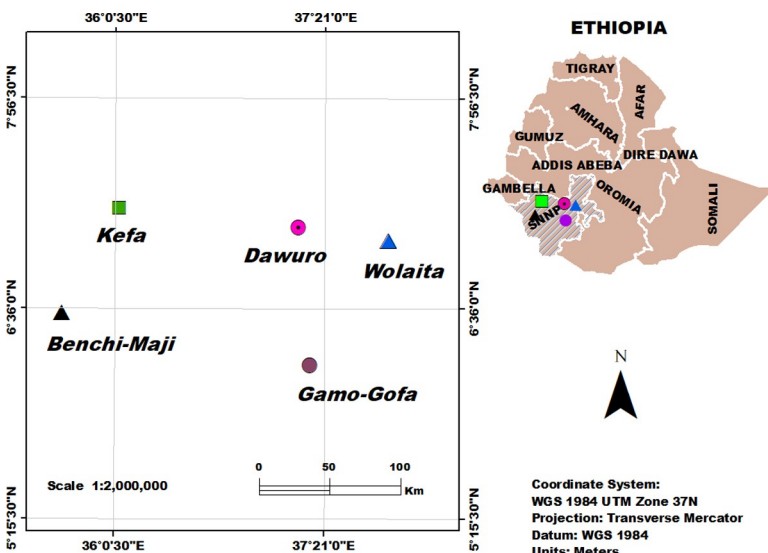

**Fig 1. Map showing administrative zones where the cocoyam accessions were collected.** The map was prepared with ArcGIS Desktop (ArcGIS Desktop 10.2.2. Esri).

sized cormels were harvested at the end of the 12<sup>th</sup> month after the planting date. The experiment was repeated in the 2017 cropping season using the same experimental design and using similar crop management practices.

## 2.3 Morphological traits and data collection

Twenty-nine morphological traits (S2 Table) were selected from the descriptors for *Xanthosoma* prepared by the International Board for Plant Genetic Resources [34]. The qualitative data were scored for qualitative traits such as plant growth habit, petiole attachment, petiole color (upper 2/3<sup>rd</sup>), petiole color (lower 1/3<sup>rd</sup>), color of the edge of petiole, lamina orientation, leaf margin color, leaf shape, color of upper leaf surface, color of lower leaf surface, position of cormel apex, shape of cormels, color of cormel apex and fresh cormel color at the end of the 7<sup>th</sup> month after the planting date, from the middle individual plant. The Munsell Plant Tissue Color Chart [35] was used to discriminate colors. The quantitative data were recorded from the middle three plants of each plot, leaving the two outside plants as border plants. The aboveground quantitative traits such as overall plant height (cm), petiole length (cm), petiole sheath length (cm), circumference of the pseudo-stem (cm), lamina length (cm) and lamina width (cm) were measured at the end of the 7<sup>th</sup> month after the planting date. The underground quantitative traits such as the number of cormels/plant, cormel length (cm), cormel diameter (cm), cormel fresh weight/plant (kg), corm length (cm), corm diameter (cm) and corm fresh weight/plant (kg) were recorded at the end of 11<sup>th</sup> months after date of plant.

## 2.4 DNA extraction

Young leaf samples were collected by using silica gel (Bio lab). Total genomic DNA was extracted using DNeasy plant min kit (QIAGEN) according to the manufacture's instruction. The DNA was checked on a 2% agarose gel. The quantity and quality of DNA were assessed by using Nanodrop-2000.

## 2.5 SSR primers and PCR amplification

Seventeen SSR primer pairs identified by Cathebras et al. [36] for *X. sagittifolium* were tested with 5 randomly selected samples. Twelve SSR primer pairs (S3 Table), which gave an intense band on an agarose gel, were selected for subsequent analysis. The forward primer of each of the 12 pairs was labeled either with 6-FAM (6-Carboxyfluorescein) or Hexachloro-fluorescein (HEX). The markers were paired into 6 PCR sets, each set containing 6-FAM labeled primer pair and HEX labeled primer pair. These labeled primers were used to test 15 samples in a pre-analysis.

PCR was performed in a total volume of 20 μL containing 2 μL of 10x PCR buffer (15 mM $MgCl_2$ included), 0.4 μL dNTP mix (10 mM each), 0.5 μL (16 pmol/μL) of each of the forward and reverse SSR primers, 2 μL genomic DNA (20 ng/μL), 0.2 μL Taq polymerase and 14.4 μL $ddH_2O$. Amplification was performed in an ABI 2720 Thermocyler. After initial denaturation at 95˚C for 2 min, 30 cycles denaturation at 94˚C for 30 s, annealing at 52˚C for 45 s, extension at 72˚C for 1 min followed with a final extension of 72˚C for 10 min were used. After verification of PCR amplification through 1% agarose gel electrophoresis, capillary electrophoresis was run by mixing the aliquots of PCR product (1 μL/well) with formamide (12 μL/well) and GeneScan™ 400HD ROX™ Size Standard (0.4 μL/well) in a 96 plate. The contents were denatured using a Boekel dry bath, which was heated to 95˚C for 5 min. Then, the DNA was quenched at -20˚C for 5 min and electrophoresed on an ABI 3130XL Genetic Analyzer (Applied Biosytems). The number and sizes of electrophoresed DNA fragments were identified using GeneMapper version 4.0 [37].

## 2.6 Data analysis

Frequency distribution of qualitative traits and basic statistical parameters based on quantitative morphological traits were calculated using SPSS version 23 [38]. Analysis of variance (ANOVA) was computed by the PROC GLM procedure in SAS 9.3 [39] because the relative efficiency of the lattice design compared with CRBD was low (less than 105%). The number of alleles (Na), the number of effective alleles (Ne) [40], Shanon information index (I) [41], and observed (Ho) and expected (He) heterozygosity within individual populations were generated using the program POPGENE version 1.32 [42]. Analysis of molecular variance (AMOVA) was performed to evaluate the partitioning of molecular variation within and among populations using GenAlEx version 6.503 [43].

Clustering, which is an important tool for visualization of variation was conducted for morphological as well as SSR marker data. The quantitative morphological traits were subjected to a cluster analysis using Minitab17.1 [44] by employing average linkage clustering strategy of the observation. Variables were standardized to a common scale by subtracting the means and dividing by the standard deviation. The number of clusters was determined by following the steps recommended by Minitab 17.1 [44]. A dendrogram showing the Euclidean distance between clusters was constructed by plotting the results of cluster analysis using the same program. The genetic clustering based on SSR data was assessed by performing the neighbor joining algorithm implemented in the computer program PHYLIP version 3.6 [45] using Nei's genetic distance [46] based on the frequencies generated by using MSA version 4.05 [47].

## 3. Results

### 3.1 Qualitative traits

Of 16 qualitative traits, 9 discriminated the cocoyam accessions into two groups, where 35 purple cocoyam accesions were grouped together and 65 green ccocoyam accessions occupied another group.Two classes of petiole color (upper 2/3$^{rd}$), petiole color (lower 1/3$^{rd}$), color of edge of petiole sheath, lamina orientation, color of upper leaf surface, color of lower leaf surface, color of veins on lower leaf surface, position of color apex and shape of cormels existed in the accessions. Qualitative traits such as plant growth habit, petiole attachment, leaf shape, leaf margin color, color of veins on upper leaf surface, color of cormel apex and fresh cormel color did not discriminate the accessions included in this study (Table 1).

### 3.2 Descriptive statistical parameters and variance

Descriptive statistics showed that the mean values of the above-ground quantitative traits such plant height, petiole length (cm) petiole sheath length (cm), lamina length (cm) lamina width (cm) and circumference of pseudo-stem (cm) were 71.49, 55.22, 31.24, 37.77, 22.02 and 28.83, respectively. The mean values of the underground quantitative traits such as the number of cormels/plant, cormel length (cm), cormel diameter (cm), cormel fresh weight/plant (kg), corm length (cm), corm diameter (cm) and corm fresh weight/plant (kg), respectively, were 10.05, 9.05, 3.26, 1.28, 11.94, 5.70 and 1.18. Maximum SD (5.63) corresponded to plant height and minimum SD (0.31) corresponded to cormel fresh weight/plant (kg). The coefficient of variation (CV %) varied from 7.65% for lamina width to 29.10% for the corm fresh weight/ plant. Analysis of variance (ANOVA) revealed significant variation in 11(84.6%) of the 13 studied quantitative traits (Tables 2 and S4).

**Table 1. Frequency distribution of 16 qualitative traits of cocoyam accessions.**

| No | Plant character | Character state | %* |
|---|---|---|---|
| 1 | Plant growth habit | Acaulescent | 100 |
| 2 | Petiole attachment | Non-peltate | 100 |
| 3 | Petiole color (upper 2/3rd) | Green | 65 |
|  |  | Purple | 35 |
| 4 | Petiole color (lower 1/3rd) | Green streaked with purple | 65 |
|  |  | Purple | 35 |
| 5 | Color of edge of petiole sheath | The same as the rest of petiole and sheath | 35 |
|  |  | Purple | 65 |
| 6 | Lamina orientation | One plane—apex down (Droopy) | 35 |
|  |  | 3-dimentional (cup-shaped) | 65 |
| 7 | Leaf shape | Sagitate basal lobes | 100 |
| 8 | Leaf margin color | Purple edge | 100 |
| 9 | Color of upper leaf surface | Medium green | 65 |
|  |  | Dark green | 35 |
| 10 | Color of lower leaf surface | Light green | 65 |
|  |  | Purplish green | 35 |
| 11 | Color of veins on upper leaf surface | Lighter green than lamina | 100 |
| 12 | Color of veins on lower leaf surface | Same as color as lamina | 65 |
|  |  | Purple | 35 |
| 13 | Position of color apex | Underground | 65 |
|  |  | Above ground and under ground | 35 |
| 14 | Shape of cormels | Globose | 65 |
|  |  | Ovate | 35 |
| 15 | Color of cormel apex | Red | 100 |
| 16 | Internal color of cormels | White streaked with purple | 100 |

*100% indicates the character common to both green leaf/petiole-cocoyam and purple leaf petiole-cocoyam accesions, 65% character specific to green- cocoyam accessions and 35% character specific to purple- cocoyam accessions.

### 3.3 Genetic diversity of cocoyam as revealed by SSR markers

Among a total of 12 SSR loci, 11 (91.7%) showed polymorphism. A total of 36 alleles were detected, ranging from 2 to 5 (average 3.273) alleles per locus with an (Table 3). The Ne per locus ranged from 1.412 (locus mXsCIR07) to 3.759 (locus mXsCIR22) with mean Ne of 2.516. The observed heterozygosity (Ho) ranged from 0.000 (loci mXsCIR19, mXsCIR16 and mXsCIR24) to 1.000 (loci mXsCIR22 and mXsCIR27) with mean Ho of 0.508. The expected heterozygosity (He) ranged from 0.294 (locus mXsCIR07) to 0.733 (locus mXsCIR22) with mean He of 0.566.

### 3.4 Genetic diversity within populations

The observed number of alleles (Na) varied from 1.727 (Bench-Maji, Kefa) to 3.091 (Gamo-Gofa) and the effective number of alleles (Ne) varied from 1.560 (Kefa) to 2.660 (Gamo-Gofa), with mean Na and Ne across populations were 2.455 and 2.091, respectively. The expected heterozygosity (He) values ranged from 0.267 (Bench-Maji) to 0.608 (Gamo-Gofa), mean He was 0.443. Shannon diversity index reflected a similar trend of He (Table 4). The partitioning of molecular variance (AMOVA) showed that the highest

**Table 2. Basic statistics and summary of mean squares of quantitative traits.**

| Quantitative traits | Mean | SD | Minimum | Maximum | CV% | Mean squares | | |
|---|---|---|---|---|---|---|---|---|
| | | | | | | Rep (df = 1) | Accessions (df = 99) | Error (df = 81) |
| PH | 71.49 | 5.63 | 57.58 | 84.83 | 7.87 | 100.30 | 82.46* | 54.46 |
| PL | 55.22 | 4.82 | 42.75 | 67.83 | 8.72 | 294.64 | 49.57* | 31.66 |
| PSL | 31.24 | 3.46 | 25.50 | 43.92 | 11.08 | 333.62 | 28.16 | 21.48 |
| LL | 37.77 | 3.20 | 29.69 | 45.25 | 8.48 | 20.09 | 16.73*** | 8.10 |
| LW | 22.02 | 1.68 | 17.28 | 25.47 | 7.65 | 0.74 | 4.96*** | 2.41 |
| CPS | 28.83 | 2.78 | 21.42 | 34.42 | 9.66 | 128.27 | 14.02*** | 5.72 |
| CN | 10.05 | 1.81 | 5.92 | 15.33 | 17.98 | 5.01 | 7.40** | 3.91 |
| CL | 9.05 | 1.09 | 6.58 | 11.83 | 12.00 | 0.02 | 2.17 | 2.30 |
| CD | 3.26 | 0.46 | 2.25 | 4.21 | 13.97 | 1.12 | 0.29** | 0.17 |
| CFW | 1.28 | 0.31 | 0.68 | 2.42 | 23.79 | 1.71 | 0.23** | 0.13 |
| CrL | 11.94 | 2.11 | 3.49 | 16.92 | 17.71 | 102.84 | 7.65** | 4.38 |
| CrD | 5.70 | 0.77 | 3.49 | 7.73 | 13.52 | 2.95 | 1.06*** | 0.45 |
| CrFW | 1.18 | 0.34 | 0.53 | 2.30 | 29.10 | 1.98 | 0.24*** | 0.12 |

PH- plant height (cm), PL-petiole length (cm), PSL-petiole heath length (cm), LL-lamina length (cm), LW-lamina width (cm), CPS-circumference of pseudo-stem (cm), CN- cormel Number per plant, CL-cormel length (cm), Cormel Diameter (cm), CFW-cormel fresh weight per plant (kg), CrL-corm length (cm), CrD-corm diameter (cm), CrFW-corm fresh weight/plant(kg), df—degree of freedom, *, ** and *** significant at p = 0.05, 0.01 and 0.001, respectively.

genetic variability (68%) existed within individuals and the lowest variability (14%) existed among populations (Table 5).

## 3.5 Clustering of accessions

The cluster analysis based on the mean values of 13 quantitative morphological traits grouped the 100 cocoyam accessions into 4 clusters (Fig 2). The clustering means of accessions based on 13 quantitative traits is presented in Table 6. Cluster four (C-IV) was the largest cluster composed of 41 (41%) of the accessions. It was also the least compact of the four clusters, having the largest within-cluster sum of squares. Accessions in this cluster

**Table 3. Genetic diversity parameters in 100 cocoyam accessions.**

| Locus name | Na | Ne | I | Ho | He |
|---|---|---|---|---|---|
| mXsCIR05 | 4 | 3.499 | 1.304 | 0.800 | 0.720 |
| mXsCIR07 | 2 | 1.412 | 0.468 | 0.355 | 0.294 |
| mXsCIR19 | 2 | 1.851 | 0.652 | 0.000 | 0.462 |
| mXsCIR10 | 4 | 2.810 | 1.087 | 0.905 | 0.648 |
| mXsCIR21 | 4 | 2.498 | 1.098 | 0.372 | 0.603 |
| mXsCIR22 | 4 | 3.759 | 1.354 | 1.000 | 0.738 |
| mXsCIR11 | 4 | 2.158 | 0.969 | 0.650 | 0.539 |
| mXsCIR24 | 2 | 1.821 | 0.643 | 0.000 | 0.453 |
| mXsCIR12 | 3 | 2.368 | 0.972 | 0.505 | 0.581 |
| mXsCIR27 | 5 | 3.689 | 1.364 | 1.000 | 0.733 |
| mXsCIR16 | 2 | 1.807 | 0.639 | 0.000 | 0.449 |
| Mean | 3.273 | 2.516 | 0.959 | 0.508 | 0.566 |
| SD | 1.103 | 0.823 | 0.318 | 0.395 | 0.142 |

Na—number of different alleles, Ne—effective number of alleles, I—Shannon information index, Ho- observed heterozygosity and He—expected heterozygosity.

**Table 4. Genetic diversity within populations.**

| Population | N | Na | Ne | Ho | He | I |
|---|---|---|---|---|---|---|
| Bench-Maji | 14 | 1.727 | 1.631 | 0.446 | 0.267 | 0.391 |
| Kefa | 12 | 1.727 | 1.560 | 0.515 | 0.289 | 0.399 |
| Dawuro | 28 | 2.818 | 2.093 | 0.492 | 0.476 | 0.787 |
| Wolaita | 28 | 2.909 | 2.513 | 0.520 | 0.574 | 0.935 |
| Gamo-Gofa | 18 | 3.091 | 2.660 | 0.540 | 0.608 | 0.990 |
| Average across populations | 20 | 2.455 | 2.091 | 0.503 | 0.443 | 0.700 |
| At species level | 100 | 3.273 | 2.516 | 0.508 | 0.566 | 0.959 |

N—number of accessions; Na—number of different alleles; Ne—effective number of alleles; Ho- observed heterozygosity; He—expected heterozygosity; I—Shannon information index.

were characterized by the highest mean value for number of cormels per plant. Cluster three (C-III) was the second largest cluster containing 27 (27%) of the total accessions. Of the 27 accessions grouped in this cluster, 25 (92.3%) were purple- cocoyam accessions. Cluster two (C-II) included the smallest number of accessions (11%) and is a more compact cluster, with the smallest within-cluster sum of squares. Accessions in this cluster were characterized by the highest mean values for plant height, petiole length, lamina length and width, circumference of pseudo-stem and corm length. Cluster one (C-I) contained 21 (21%) accessions, which were characterized by the lowest mean values for most of the traits studied. The cluster analysis showed that accessions collected from different zones (populations) were clustered together. Clustering individual accessions based on SSR data is presented in the neighbor-joining (NJ) tree constructed using on Nei's genetic distances. The clustering fully separated green- cocoyam accessions from the purple- cocoyam accessions (Fig 3).

## 4. Discussion

This study employed the techniques of genetic characterization using morphological traits and SSR molecular markers, which provided insight into a broader context for the genetic diversity and clustering of cocoyam accessions from Ethiopia. The majority (84.6%) of the quantitative morphological traits showed significant variation among accessions. Similar to this finding other authors, who had assessed genetic diversity of cocoyam accession by using morphological traits also reported significant variation among the studied accessions [24, 48, 49]. SSR markers revealed high genetic diversity across populations. Most of (91.7%) the SSR loci were polymorphic within cocoyam accessions. In a related study, Doungous et al. [50] reported 88% to 97% polymorphic loci across 20 cocoyam accessions from Cameroon using retrotransposon molecular markers. The average He across populations (He = 0.443) was higher than that of *Ambrosina bassii* L. (Araceae, Ambrosineae) (He = 0.263) [51] and *Anthurium crenatum* (L.) Kunth (Araceae) (He = 0.167) [52],

**Table 5. Summary of AMOVA for five cocoyam populations based on SSR markers.**

| Source of variation | df | Sum of squares | Estimated variance | % of variation | Fst | P value |
|---|---|---|---|---|---|---|
| Among Pops | 4 | 94.046 | 0.510 | 14 | 0.143 | 0.001 |
| Among Indivs | 95 | 349.994 | 0.637 | 18 | | |
| Within Indivs | 100 | 241.000 | 2.410 | 68 | | |
| Total | 199 | 685.040 | 3.558 | | | |

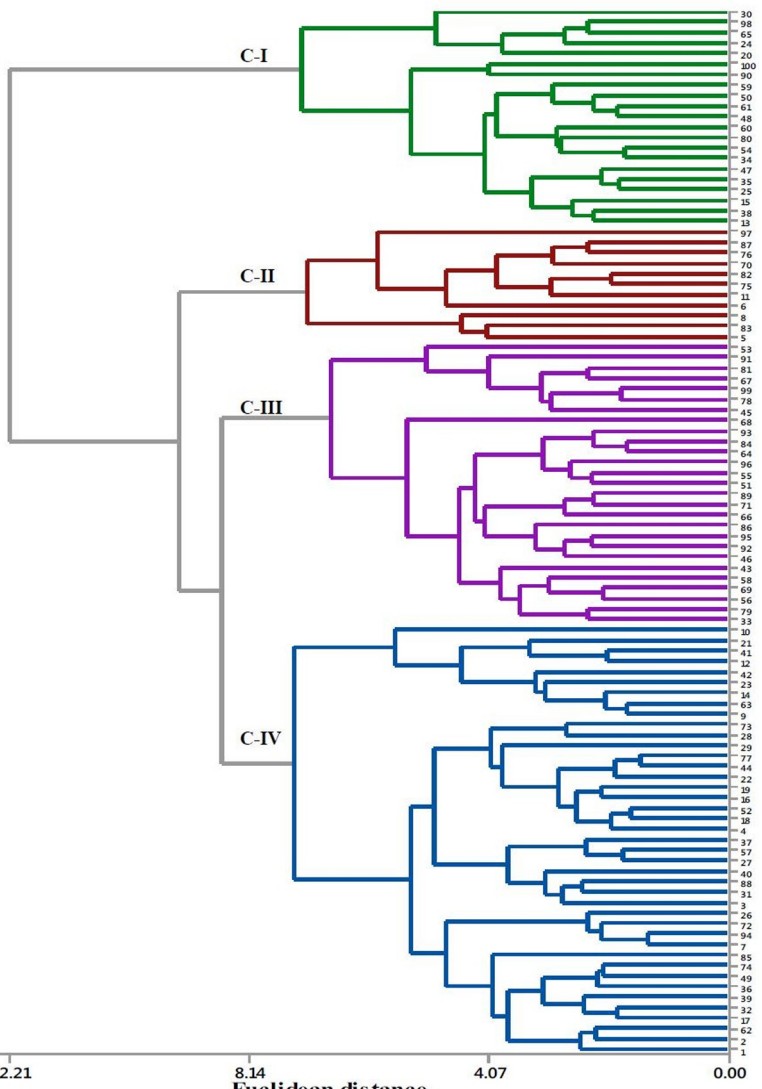

**Fig 2. Cluster analysis showing the relationship among cocoyam accessions based on quantitative traits.** The numbers stand for the serial numbers of the accession code. The accessions 1–14, 15–26, 27–54, 55–82 and 83–100 were from Bench-maji, Kefa, Dawuro, Wolaita and Gamo-Gofa zones, respectively.

implying that our collections contained genetically diverse cocoyam accessions. The observed heterozygosity was very high (Ho = 1) for two loci (mXsCIR22 and mXsCIR27) (Table 3) and six SSR markers presented Ho values higher than He values. SSR markers identified by Cathebras et al. [36], to assess cocoyam genetic diversity, gave variable degrees of heterozygosity, observed at levels ranging from 0.000 to 0.970. This excess in heterozygotes might be due to high rates of asexual reproduction in cocoyam due to vegetative propagation.

The results of genetic clustering analyses have a range of applications, including, conservation, breeding, and understanding crop evolution [53]. The qualititative morphological traits and SSR markers-based clustering analyses effectively separated the purple cocoyam accessions from green cocoyam accessions. Cluster analysis based on quantitative morphological data also grouped most of (92.3%) the purple cocoyam accessions in

**Table 6. Cluster means of 13 quantitative traits of cocoyam.**

| Quantitative traits | Means values of cluster | | | |
|---|---|---|---|---|
| | C-I | C-II | C-III | C-IV |
| PH | 67.26 | 80.00 | 67.60 | 73.93 |
| PL | 50.16 | 63.16 | 55.15 | 55.51 |
| PSL | 38.44 | 37.86 | 30.80 | 31.21 |
| LL | 34.50 | 41.13 | 39.71 | 37.28 |
| LW | 20.33 | 23.89 | 22.63 | 22.14 |
| CPS | 26.00 | 31.69 | 30.42 | 28.48 |
| CN | 9.19 | 10.32 | 9.28 | 10.93 |
| CL | 9.03 | 9.61 | 8.60 | 9.21 |
| CD | 2.91 | 3.59 | 3.68 | 3.08 |
| CFW | 1.02 | 1.43 | 1.31 | 1.35 |
| CrL | 10.27 | 12.30 | 14.08 | 11.28 |
| CrD | 5.00 | 6.19 | 6.31 | 5.55 |
| CrFW | 0.92 | 1.29 | 1.51 | 1.07 |
| Within cluster sum of squares | 167.63 | 117.39 | 192.94 | 304.46 |

PH- plant height (cm), PL-petiole length (cm), PSL-petiole heath length (cm), LL-lamina length (cm), LW-lamina width (cm), CPS-circumference of pseudo-stem (cm), CN- cormel Number per plant, CL-cormel length (cm), Cormel Diameter (cm), CFW-cormel fresh weight per plant (kg), CrL-corm length (cm), CrD-corm diameter (cm) and CrFW-corm fresh weight/plant(kg).

distinct cluster (cluster III), suggesting the existence of genetic difference between green- and purple-cocoyam accessions. The color difference could be considered as important for easy identification of accessions. Most of the accessions collected from different zones (populations) were collected from different zones, were grouped regardless of the collection zones, indicating that there has been movement of germplasm among zones, possibly by farmer-to-farmer planting materials exchange. Previously, a similar study [26] showed that most of the cocoyam accessions from different districts and villages of Ethiopia were clustered together.

## 5. Conclusion

The exact date and source of cocoyam introduction into Ethiopia have not been definitely determined. The morphological and molecular genetic diversity analysis conducted in this study showed high variation in morphological traits and genetic diversity, the latter shown by the total number of alleles (36) with an average of 3.273 alleles per locus coupled with high average observed heterozygosity (0.503) and expected heterozygosity (0.443) values across populations. Despite being a relatively recent introduction to Ethiopia, our results suggest that geneticaly diverse cocoyam accessions representing multiple lineages may have been introduced into the country multiple times, through multiple routes and probably by multiple agents, a hypothesis that needs further testing and analysis. Furthermore, the crop has quickly dispersed and gained popularity among farmers on account of its usefulness and adaptability to the agroclimatic zones and socio-cultural settings of its current occurrence/distribution. The crop, therefore, needs more research efforts commensurate with its economic and social values than it has been accorded thus far. Further study is recommended to clarify the taxonomic status of Ethiopian cocoyam accessions and to trace their evolutionary relationship with *Xanthosoma* species elsewhere.

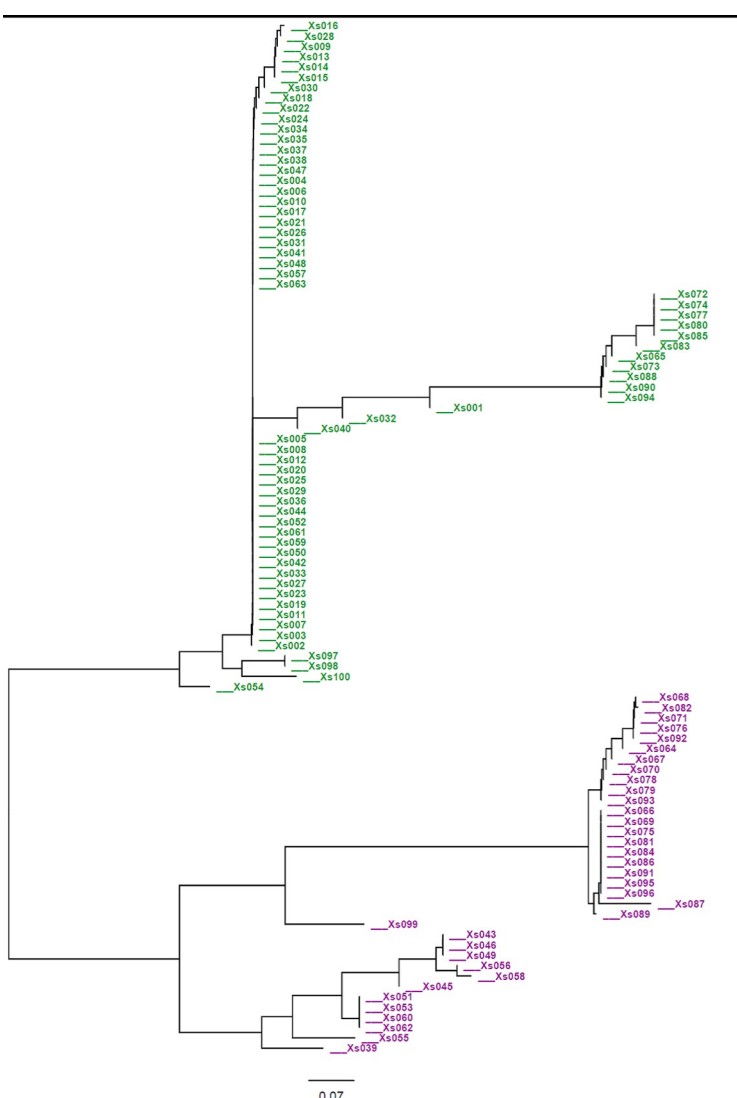

**Fig 3. Neighbour joining tree showing the relationships among Ethiopian cocoyam accessions.** The tree was generated based on Nei's genetic distance using 11 microsatellite markers. Green- and purple- represent green- and purple- cocoyam accessions. The accessions Xs001-Xs014, Xs015-Xs026, Xs027-Xs054, Xs055-Xs082 Wolaita and Xs083—Xs100 were from Bench-maji, Kefa, Dawuro, Wolaita and Gamo-Gofa zones, respectively.

## Supporting information

**S1 Table. Passport data.**
(DOCX)

**S2 Table. Selected morphological descriptors.**
(DOCX)

**S3 Table. Primers sequences.**
(DOCX)

**S4 Table. Mean performances of 13 quantitative traits of 100 cocoyam accessions.**
(DOCX)

## Acknowledgments

The authors appreciate Dr. Asfaw Kifle for the assistance during the field works and during morphological data collection. Malli Aradhya and Gloria Gloria Diaz-Britz (USDA National Clonal Germplasm Repository, Davis) provided generous support and access to equipment for the SSR analyses. Michael Mears assissted with laboratory procedures. Wolaita Sodo University provided vehicle for germplasm collection and the field experiment was conducted at Areka Agricultural Research Center, Ethiopia.

## Author Contributions

**Conceptualization:** Eyasu Wada, Tileye Feyissa, Kassahun Tesfaye, Zemede Asfaw.

**Formal analysis:** Eyasu Wada, Daniel Potter.

**Funding acquisition:** Eyasu Wada, Daniel Potter.

**Investigation:** Eyasu Wada, Tileye Feyissa, Kassahun Tesfaye, Zemede Asfaw, Daniel Potter.

**Writing – original draft:** Eyasu Wada.

**Writing – review & editing:** Eyasu Wada, Tileye Feyissa, Kassahun Tesfaye, Zemede Asfaw, Daniel Potter.

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
