## [Decision Letter · Decision Letter 0]

26 Aug 2020

PONE-D-20-24312

Genetic diversity of Ethiopian cocoyam (Xanthosoma sagittifolium (L.) Schott) accessions as revealed by morphological traits and SSR markers

PLOS ONE

Dear Dr. Wada,

Thank you for submitting your manuscript to PLOS ONE. After careful consideration, we feel that it has merit but does not fully meet PLOS ONE’s publication criteria as it currently stands. Therefore, we invite you to submit a revised version of the manuscript that addresses the points raised during the review process.

We look forward to receiving your revised manuscript.

Kind regards,

Tzen-Yuh Chiang

Academic Editor

PLOS ONE

Journal Requirements:

"This study was supported by a grant from the DAAD, Wolaita Sodo University and

Department of Plant Sciences of University of California-Davis. We thank Areka

Agricultural Research Center for experimental field and USDA for SSR fragment analysis.

We appreciate the help of Dr. Asfaw Kifle during collection of accessions and during field

works."

Reviewers' comments:

Reviewer's Responses to Questions

**Comments to the Author**

1. Is the manuscript technically sound, and do the data support the conclusions?

Reviewer #1: Partly

Reviewer #2: Partly

2. Has the statistical analysis been performed appropriately and rigorously? 

Reviewer #1: Yes

Reviewer #2: No

3. Have the authors made all data underlying the findings in their manuscript fully available?

Reviewer #1: No

Reviewer #2: Yes

4. Is the manuscript presented in an intelligible fashion and written in standard English?

Reviewer #1: Yes

Reviewer #2: No

5. Review Comments to the Author

Reviewer #1: The genus Xanthosoma includes three important food species: X. sagittifolium, X. atrovirens and X. violaceum. Another species, X. brasiliense, is grown for its edible leaves. Xanthosoma sagittifolium is a robust plant reaching heights of 2 m or more. The major difference between it and taro is in leaf shape. Xanthosoma spp. have sagittate leaves, while Colocasia spp. have peltate leaves. There is a sentence in the introduction explaining that in Ethiopia the two are considered synonyms. This is very confusing because the reader has the feeling that while collectiong germplasm, the authors might have collected Colocasia as well. Leaves of Xanthosoma are thick and long-petioled, with the main vein at the lower side of either basal lobe being marginal as it joins the petiole. The leaves are arrow-shaped with sharp tips and deep, wide basal lobes and a prominent marginal vein. Leaf petioles can be more than 2 m long, with blades more than 1 m long and up to 0.7 m wide. Usually the corm of such plant is about 10 to 15 cm in diameter but here diameter of 5-6 cm are reported which is shedding doubts about the species involved. The authors need to provide a table with the passport data and the information on the voucher spécimens deposited in their National Herbarium. This is common practice when diversity studies are conducted to avoid confusion and possible doubts about the Identity of the material analysed.

But the paper lacks a scientific question. At the moment this is a diversity study of an american species (only one or several Xanthosoma sp?) introduced in Ethiopia using molecular markers. It appears that there is significant variation but there is no explanation for it, except that different genotypes may have been introduced.

The authors are therefore invited to:

Clarify their objective, elaborate on the hypotheses they want to test

Provide more details and passport data on their material

Clarify the identity of their material, is only one Xanthosoma species involved or several?

provide a map indicating where the accessions were collected

improve the quality of the dendrograms, the figures are blurry and difficult to read

Reviewer #2: In this study 100 cocoyam accessions were collected from five regions in Ehtiopia, after which they were multiplied and then planted in a single trial which was used for descriptive characteristics and SSR analysis. The environment as well as season has a highly significant influence on the expression of plant characteristics. A plant could look very different when grown in two environments and when evaluated in different seasons. Therefore to do any meaningful characterization of morphological traits in order to determine genetic diversity, multiple environments and seasons should be included in the study, otherwise the data is simply nor reliable.

Other comments:

One assumes from the tables that two replications were used in the field trial? This should be described more clearly.

6. PLOS authors have the option to publish the peer review history of their article (what does this mean?). If published, this will include your full peer review and any attached files.

Reviewer #1: **Yes: **Vincent Lebot

Reviewer #2: No

---

## [Author Response · Author response to Decision Letter 0]

16 Oct 2020

Response to Reviewers

First we would like to thank the reviewers for taking their time to give us these constructive comments. We have replied to the comments as follows:

• Response to general comments 

o Our manuscript have been carefully checked and revised as needed according to the PLOS ONE’s style requirements.

o The financial disclosure statement is changed and the cover letter is updated

o Figure files are resubmitted carefully checking PLOS ONE requirements of figure files 

o ORCID included

o Captions for supporting information files are included at the end of the manuscript and in text citations are updated 

Reply to reviewer’s comment 

Reviewer #1

• Clarify objective and elaborate hypothesis.

o Objective is clarified and hypothesis is elaborated 

• Provide passport data? 

o Provided as supplementary information (S1 Table. Passport data)

• Provide Map? 

o Provided

• Improve Quality of dendograms and figures?

o Improved 

• A reader may be confused about our study materials and feel that Colocasia might have been included because of a sentence in the introduction explaining that in Ethiopia the two species are considered synonymous.

o We have edited the sentence and clarified it to avoid any confusion

• Size and diameter of corm?

o The size and diameter of Xanthosoma corm is 10 to 15 cm in diameter but here the average diameter of corm recorded from 3 replication for two years experiment were 4.29 to 7.73 cm and the overall mean was 5.70 cm. Such difference may be due to: 

1) Cocoyam is a perennial plant, but for practical purposes, the corm and cormles were intentionally measured separately at the end of 12th months after plantation (annually). This annual harvest could be the reason for the size and diameter reduction. Actually, we encountered larger sized corm at some fields where cocoyam grown as perennial crop when we were collecting germplasms, but which was not at research field (where it was cultivated annually).

2) The soil type of the experimental site could be another reason.

• One or several Xanthosoma species?

o In general, the division of the genus Xanthosoma into species has been difficult [1]. There is much confusion, discrepancies and uncertainties regarding the taxonomy of Xanthosoma at the species level. Various names have been used synonymously [2-3], the same plant being given more than one species name [4]. Thus, the name of Xanthosoma sagittifolium (L.) Schott has usually been given to the most cultivated members of this genus (Giacometti and [5-6]. In the course of this study, we refer to the accepted species status of X. sagittifolium in The Plant List [2] in the Araceae family [6], which does not tackle taxonomy in further detail.

o There was no substantial evidence to say that there were different Xanthosoma species in Ethiopia although qualitative morphological traits and SSR markers clustered accession into two groups. We have a plan to do taxonomic study on Xanthosoma species in Ethiopia. Generally, the taxonomy of Xanthosoma needs to be reconstructed. 

1. Saborio, F. (2007). Cocoyam (Xanthosoma sagittifolium (L.) Schott). In: Breeding of Neglected and Underutilized Crops, Spices and Herbs, pp. 172-189, (Ochatt, S. and Jain, S.M., eds). Science Publishers, USA. 

2. The Plant List (http://www.theplantlist.org) 

3. Lim, T.K. (2015). Edible Medicinal and Non-Medicinal Plants. Volume 9: Modified Stems, Roots, Bulbs. New York: Springer.

4. O’Hair, S.K. and Maynard, D.N. (2003). Edible aroids. In: Encyclopedia of Food Science and Nutrition, PP. 5970-5973, (Trugo, L.C. Finglas, P.M. Belton, P. Ottaway, P.B., Bressani, R., et al., eds). Academic press, USA.

5. Giacometti, D.C. and Leon, J. (1994). Tannia, yautia (Xanthosoma sagittifolium). In: Neglected Crops 1492 from Different Perspectives, pp. 255–258, (Hernandez-Bermejo, L., ed). Food and Agricultural Organization of United the United Nations, Rome, Italy.

6. Govaerts, R., Frodin, D.G. and Bogner, J. (2002). World checklist and bibliography of Araceae (and Aroraceae). Royal Botanic Gardens, UK.

Reviewer #2

• Design and Replication? 

o Similar sized cormel germplasms were sampled from five Zones. Then, characterized using morphological traits by planting in a common research garden in 10x10 simple lattice square design for two years. This is now clarified in the methodology. 

o The genotype by environmental interaction (GXE) evaluation is the next plan to be done. Therefore, multilocational trials is to be conducted in different agroecology to assess the magnitude of genotype x environment (GXE) interaction and to identify stable genotypes and evaluate their performance, selecting some genotypes from the clustered accessions to evaluated yield and yield related traits.

Reviewer #3

Data availability 

o All data made to be available either as part of manuscript in table or as supplementary materials. This is also preferred comment from other two reviewers 

Manuscript presentation

o Some improvements were made on the presentation of the paper as needed

Objective and Hypothesis 

o Objective is clarified and hypothesis is elaborated 

Sample collection zones

o The accessions that were collected from the same zone were considered as one population for the genetic diversity studies in assumption that cormel germplasms were more likely shared within zones than among zones. The same language is spoken within the zones and there are frequent local markets within zones where local people buy and sell commodities. This is added in the methods and material section for clarity 

Results

3.1 Start the result section with clear finding

o Edited to meet the reviver’s comment

3.2 Please don’t start a sentences with a Table 

o The comment is accepted and corrected 

I would combine Table 2 and Table 3, emphasis to statistical significant character

o The comment accepted and the two tabled combined. Emphasis was given to statistical significant character, where statistical comparison was made at p= 0.05, 0.01 and 0.001

3.3 Please mention the link to Table 4 early in the text

o The link to table 4 is mentioned early in the text 

Please check data and recalculate data for Ho =1 

o Data checked and recalculated for Ho =1 for two loci (mXsCIR22 and mXsCIR27) (Table 3) and six SSR markers presented Ho values higher than He values. SSR markers, which we have used were initially identified by Cathebras et al. [1], gave variable degrees of heterozygosity, observed at levels ranging from 0.00 to 0.97. This excess in heterozygotes might be due to high rates of asexual reproduction in cocoyam due to vegetative propagation. The inflorescence of cocoyam is protogynous [2].

1. Catherbras C. Traore C. Malapa R. Risterucci A. Chair H. characterization of microsatellites in Xanthosoma sagittifolium (Araceae) and cross-amplification in related species. Appl Plant Sci. 2014; 2(6); 2(6):129-134 http://doi.org/10.3732/apps.1400027.

2. Manner HI. Farm and forestry production and marketing profile for Tannia (Xanthosoma spp.). In: Elevitch CR, editor. Specialty Crops for Pacific Island Agroforestry. Permanent Agriculture Resources, Hawaii, USA; 2011. pp. 1-16.

How many samples from each population 

o Twelve to twenty eight accessions were sample from each zone. We clarified this now in the methodology section and the accessions passport data is presented as supplementary information (S1 Table 1) according the reviewer #1 and Reviewer #2 comments 

3.4 Don’t repeat the values in the Table, extract the most important findings 

o Comment accepted accordingly

3.5 is not worth a separate chapter. Please combine with the previous

o Combined and the main finding is interpreted at the discussion section 

3.6 Is this clustering of accessions separate from the morphological analysis?

o No, clusters were done by using both morphological and SSR markers data. The results were presented by Fig. 2 and Fig. 3 to see how one goes with the other. It was seen that the second larger cluster (C-III) was contained 27 (27%) of the total accessions. Of the 27 accessions grouped in this cluster, 25 (92.3%) were purple-cocoyam accessions (Fig 2). This result was obtained by using data of morphological genetic traits. Molecular data was also grouped almost all purple-cocoyam accessions in one cluster and the green ones in another cluster (Fig.2), supporting the morphological data. These results were presented in the result section and possible reasons for this were explained in the discussion section.

In M&M it was not explained why clustering seems to be important 

o Checked and explained

How clustering was conducted

o The following sentences are in the M&M which explain how clustering of accession were conducted. 

• The quantitative morphological traits were subjected to a cluster analysis using Minitab17.1, which was conducted by employing average linkage clustering strategy of the observation. Variables were standardized to a common scale by subtracting the means and dividing by the standard deviation. The number of clusters was determined by following the steps recommended by Minitab 17.1. The dendrogram showing the Euclidean distance between clusters was constructed by plotting the results of cluster analysis using the same program.

• The genetic clustering based on SSR data was assessed by performing using neighbour joining algorithm implemented in the computer program PHYLIP version 3.6 using Nei’s genetic distance based on the frequencies generated by using MSA version 4.05.

Discussion 

is too descriptive, several values that were mentioned in the result section were mentioned again, the discussion must be shortened, taxonomic conclusion need to be drawn 

o These comments were considered carefully and corrected accordingly.

---

## [Decision Letter · Decision Letter 1]

6 Nov 2020

PONE-D-20-24312R1

Genetic diversity of Ethiopian cocoyam (Xanthosoma sagittifolium (L.) Schott) accessions as revealed by morphological traits and SSR markers

PLOS ONE

Dear Dr. Wada,

Thank you for submitting your manuscript to PLOS ONE. After careful consideration, we feel that it has merit but does not fully meet PLOS ONE’s publication criteria as it currently stands. Therefore, we invite you to submit a revised version of the manuscript that addresses the points raised during the review process.

We look forward to receiving your revised manuscript.

Kind regards,

Tzen-Yuh Chiang

Academic Editor

PLOS ONE

Reviewers' comments:

Reviewer's Responses to Questions

**Comments to the Author**

1. If the authors have adequately addressed your comments raised in a previous round of review and you feel that this manuscript is now acceptable for publication, you may indicate that here to bypass the “Comments to the Author” section, enter your conflict of interest statement in the “Confidential to Editor” section, and submit your "Accept" recommendation.

Reviewer #1: All comments have been addressed

Reviewer #2: All comments have been addressed

2. Is the manuscript technically sound, and do the data support the conclusions?

Reviewer #1: Partly

Reviewer #2: Yes

3. Has the statistical analysis been performed appropriately and rigorously? 

Reviewer #1: Yes

Reviewer #2: Yes

4. Have the authors made all data underlying the findings in their manuscript fully available?

Reviewer #1: No

Reviewer #2: Yes

5. Is the manuscript presented in an intelligible fashion and written in standard English?

Reviewer #1: No

Reviewer #2: No

6. Review Comments to the Author

Reviewer #1: Xanthosoma sagittifolium is an allogamous species of American origin which does not set fertile seeds in farmers' fields in Ethiopia. Therefore, all genotypes have been clonally introduced. The authors should identify the number of multilocus genotypes (MLGs) and discuss the possible number of distinct clonal lineages before concluding that there is enough genetic diversity and the have not attempted to conduct crosses. What is the number of triploids in the introduced clones ? Triploids being useless for breeding purposes, this reduce the useful genetic diversity.

Reviewer #2: There are still numerous language and writing errors. It is strongly recommended that the paper be professionally language edited. I have tried to correct as many as possible errors in the attached pdf.

7. PLOS authors have the option to publish the peer review history of their article (what does this mean?). If published, this will include your full peer review and any attached files.

Reviewer #1: No

Reviewer #2: No

---

## [Author Response · Author response to Decision Letter 1]

25 Nov 2020

Response to Reviewers

First we would like to thank the reviewers for taking their precious time to give us constructive comments. We have replied to the comments as follows:

• Reply to reviewer’s comment 

o Our manuscript has been carefully checked per the PLOS ONE’s style requirements

Reviewer #1

1. Data availability?

o All available data, underlying the findings described in our manuscript, are provided as part of the manuscript and its supporting information files, without restriction. 

2. The authors should identify the number of multilocus genotypes (MLGs) and discuss the possible number of distinct clonal lineages before they attempt to conduct crosses.

o We have edited our conclusion and further plan according to this reviewer comment

Reviewer #1 and Reviewer #2

1. There are language and writing errors 

o We would like to thank reviewer #2 for the given corrections. We have accepted the corrections accordingly and further professionally editing was also done per the recommendation forwarded by reviewer #2

---

## [Decision Letter · Decision Letter 2]

11 Dec 2020

PONE-D-20-24312R2

Genetic diversity of Ethiopian cocoyam (Xanthosoma sagittifolium (L.) Schott) accessions as revealed by morphological traits and SSR markers

PLOS ONE

Dear Dr. Wada,

Thank you for submitting your manuscript to PLOS ONE. After careful consideration, we feel that it has merit but does not fully meet PLOS ONE’s publication criteria as it currently stands. Therefore, we invite you to submit a revised version of the manuscript that addresses the points raised during the review process.

We look forward to receiving your revised manuscript.

Kind regards,

Tzen-Yuh Chiang

Academic Editor

PLOS ONE

Reviewers' comments:

Reviewer's Responses to Questions

**Comments to the Author**

1. If the authors have adequately addressed your comments raised in a previous round of review and you feel that this manuscript is now acceptable for publication, you may indicate that here to bypass the “Comments to the Author” section, enter your conflict of interest statement in the “Confidential to Editor” section, and submit your "Accept" recommendation.

Reviewer #2: All comments have been addressed

2. Is the manuscript technically sound, and do the data support the conclusions?

Reviewer #2: Yes

3. Has the statistical analysis been performed appropriately and rigorously? 

Reviewer #2: Yes

4. Have the authors made all data underlying the findings in their manuscript fully available?

Reviewer #2: Yes

5. Is the manuscript presented in an intelligible fashion and written in standard English?

Reviewer #2: Yes

6. Review Comments to the Author

Reviewer #2: There are still some small writing and language errors that need correction. They are indicated on the attached pdf.

7. PLOS authors have the option to publish the peer review history of their article (what does this mean?). If published, this will include your full peer review and any attached files.

Reviewer #2: No

---

## [Author Response · Author response to Decision Letter 2]

19 Dec 2020

• A reviewer comment: There are still some small writing and language errors that need correction. They are indicated on the attached pdf. 

• Response: We would like to thank the reviewer for the given corrections. We have included the corrections.

---

## [Editor Report · Decision Letter 3]

23 Dec 2020

Genetic diversity of Ethiopian cocoyam (Xanthosoma sagittifolium (L.) Schott) accessions as revealed by morphological traits and SSR markers

PONE-D-20-24312R3

Dear Dr. Wada,

We’re pleased to inform you that your manuscript has been judged scientifically suitable for publication and will be formally accepted for publication once it meets all outstanding technical requirements.

Kind regards,

Tzen-Yuh Chiang

Academic Editor

PLOS ONE
---

## [Editor Report · Acceptance letter]

28 Dec 2020

PONE-D-20-24312R3 

Genetic diversity of Ethiopian cocoyam (*Xanthosoma sagittifolium* (L.) Schott) accessions as revealed by morphological traits and SSR markers 

Dear Dr. Wada:

I'm pleased to inform you that your manuscript has been deemed suitable for publication in PLOS ONE. Congratulations! Your manuscript is now with our production department. 

Kind regards, 

on behalf of

Dr. Tzen-Yuh Chiang 

Academic Editor

PLOS ONE